# Effect of the Inorganic Modification Mode on the Mechanical Properties of Rubber Recycled Concrete

**DOI:** 10.3390/ma17102217

**Published:** 2024-05-08

**Authors:** Leifei Liu, Jingmei Zong, Xueqian Hou, Xiaoyan Liu

**Affiliations:** 1School of Civil Engineering, Shangqiu Institute of Technology, Shangqiu 476000, China; liuleifei2012@126.com (L.L.); zongjingmei1987@163.com (J.Z.); 16692656070@163.com (X.H.); 2State Key Laboratory for Geomechanics and Deep Underground Engineering, China University of Mining and Technology, Xuzhou 210096, China

**Keywords:** rubber, recycled aggregate, concrete, mechanical properties, modification methods

## Abstract

The reasonable and effective application of waste tires and discarded concrete in concrete is an important branch of green concrete development. This paper investigates the effects of the inorganic modification mode on the basic mechanical properties of rubber recycled concrete based on indoor tests. Inorganic modification, such as water washing, acid washing, and alkaline washing modification, was mainly used to treat and modify rubber particles. The factors affecting the compressive strength, the splitting tensile strength, the flexural strength, the axial compressive strength, and the modulus of elasticity of modified rubber recycled concrete were analyzed. The study results show that the incorporation of recycled aggregates and rubber reduced the mechanical properties of concrete, with the compressive and splitting tensile strengths showing the greatest reductions of 27.36% and 27.24%, respectively. Three modification methods significantly improved the mechanical properties of rubber recycled concrete. The alkali washing modification method was the most effective, maximally improving the mechanical properties of rubber recycled concrete by 7.53–15.51%. The effects of the three modifications on the mechanical properties of concrete were ranked as follows: alkali washing > acid washing > water washing. This study provides a data basis for the practical application of rubber recycled concrete in engineering and a test basis for the development of green concrete.

## 1. Introduction

With the rapid development of the construction and automobile industries, a large amount of waste such as discarded concrete and used rubber tires has been generated [1,2]. Large quantities of used tires and construction waste not only take up land resources but may also cause pollution to the environment and waste resources. Modified rubber recycled concrete (MRRC) is a new type of material that combines the properties of rubber granules from waste tires with recycled building concrete materials. This solves the problem of the disposal of waste tires and waste concrete, significantly reducing the cost of production of new materials with better economic benefits [3,4,5]. By incorporating rubber granules and waste concrete into concrete making, not only can the number of waste tires be reduced, but also resources can be recycled by converting waste resources into valuable building materials [6,7]. Therefore, it is significant to investigate the green and environmentally friendly building materials of modified rubber recycled concrete (MRRC).

Research on rubber recycled concrete (RRC) mainly focuses on the following aspects: Formulation technology of rubber recycled concrete mix. Batayneh et al. (2008) [8] called for the use of waste rubber to study concrete configuration techniques, promoting the development of waste tires in concrete; Rassokhin et al. (2022) [9] studied a low-strength fine aggregate-based concrete by modern construction chemistry in combination with pozzolan and high dispersed ground additives and nanomodification; Wu et al. (2020) [10] and Hameed et al. (2023) [11] mainly investigated the effect of the compression casting technique on the mechanical properties and microstructure of rubber recycled concrete. Basic mechanical properties, such as the compressive strength, the axial compressive strength, the flexural strength, elastic modulus, and stress–strain curve were determined. This section is highlighted in the following section. Durability, such as resistance to chloride penetration, freezing and thawing, and corrosion, was investigated. Guo et al. (2017) [12], Grinys et al. (2021) [13], Amiri et al. (2021) [14], and Ataria and Wang (2022) [15] mainly investigated the effects of surface treatment methods, coating techniques, rubber particle size and content on strength, durability, and freeze–thaw resistance of RRC. The synergistic effect of waste aggregates and rubber may reduce the durability of concrete, but the NaOH treatment can enhance the durability and frost resistance of concrete. Tang et al. (2021) [16] and Tang et al. (2021) [17] investigated the mechanical properties of rubberized recycled concrete at high temperatures, such as the compressive strength, stress–strain curve, deformation performance, and bursting resistance. Feng et al. (2022) [18] further investigated the thermo-mechanical properties of RRC and proposed a thermo-mechanical property prediction model based on machine learning. Microscopic properties. He et al. (2021) [19] and Juveria et al. (2023) [20] used infrared spectroscopy to investigate the surface changes in modified rubber modified with sodium hydroxide, sulfonation, and urea. The hydration mechanism and inter-particle bonding process of rubber recycled concrete were also investigated using SEM. Shao et al. (2022) [21] investigated the effect of recycled tire rubber and carbon nanotubes on the microscopic properties of concrete, and explored the mechanism of the two mixtures in the hydration reaction of cement.

Mechanical properties are significant in the research field of modified rubber recycled concrete. The effects of the mixing of recycled aggregates, the size, type, and content of rubber, the cement content, and various fibers and other additives on the basic mechanical properties of concrete have been studied. Liu et al. (2015) [22] investigated the fatigue performance and damage characteristics of modified recycled aggregate concrete with 10–30% rubber admixture, and obtained that the fatigue life of concrete can be significantly improved for 20% rubber admixture. Further, Aslani (2016) [23], Stallings et al. (2019) [24], and Karunarathna et al. (2021) [25] supplemented the study of the rubber type, content, particle size on the mechanical properties of concrete and obtained a significant increase in the toughness of the improved concrete. Su et al. (2015) [26] investigated the effect of waste rubber tire aggregate on concrete strength, shrinkage, and water resistance. Mhaya et al. (2021) [27] evaluated the effect of waste rubber tires and blast furnace slag on the mechanical properties and impact resistance of concrete, and obtained that optimum mechanical properties of concrete were obtained for 5–30% rubber admixture. In terms of dynamic properties, Najim and Hall (2012) [28] and Li et al. (2016) [29] investigated the effects of rubber particles, particle size, admixture, and strain rate on the dynamic properties and toughness of concrete, and obtained waste-rubber-modified recycled-aggregate concrete with better mechanical properties. For modification methods, He et al. (2016) [30] and Su et al. (2022) [31] investigated the effects of physical modification and organic modification on the mechanical properties of rubber-cement concrete. Silane coupling agent (SCA) has better mechanical properties than NaOH-modified concrete. Further, the effects of different fibers and various polymer admixtures, such as crumb rubber, natural zeolite, natural rubber post-bonding material [32], nano-SiO_2_ (NS) solution and NS sol–gel [33], on the mechanical properties of concrete were studied. The effect of fiber was obtained to be greater than that of recycled aggregate and rubber aggregate admixture.

In summary, existing studies have focused on the effects of recycled aggregates and rubber admixture on the mechanical properties of concrete. However, not much research has been performed on the synergistic effect of recycled aggregates and modified rubber particles. Further research is needed to study the effect of the interaction mechanism between the two on concrete. Further, there is little research on the effect of inorganic modification methods on the mechanical properties of MRRC and this needs to be further explored. Therefore, this paper investigates the synergistic mechanism of combining recycled aggregates and rubber particles based on indoor tests. The effects of different modifications on the mechanical properties of recycled aggregate concrete were investigated. The results of this study provide some references for the application and development of recycled resources in concrete.

## 2. Materials and Methods

### 2.1. Materials

The raw materials used in this test mainly included cement, coarse aggregate, fine aggregate, water, and modifier. P.O. 42.5 ordinary silicate cement from Jiangsu Lianyungang Banqiao Zhonglian Cement Co. (Lianyungang, China), with an initial setting time of 200 min, a final setting time of 275 min, a fineness of 2.0%, and satisfactory bulk stability, was adopted. Its indicators were in line with the requirements of “*Common portland cement*” (GB175-2007 [34]). The basic performance indexes of P.O. 42.5 ordinary silicate cement are shown in Table 1.

Coarse aggregate was divided into two types: natural coarse aggregate and recycled coarse aggregate. The natural coarse aggregate was continuously graded crushed stone and the recycled coarse aggregate was made from waste concrete beams by crushing and screening. Fine aggregates included both ordinary natural river sand and rubber granules. Table 2 shows the basic performance indexes of coarse and fine aggregates. Further, the modifiers used in this test were solid NaOH and solid MgSO_4_ produced by Shanghai Runjie Chemical Reagent Science and Technology Co., Ltd. (Shanghai, China) and the modified solution was prepared according to the designed concentration. The specific materials used in this test are shown in Figure 1.

### 2.2. Test Procedure

The design strength of ordinary concrete was C40, with a water–cement ratio of 0.42 and a sand rate of 0.3. According to “*The specification for the mix proportion design of ordinary concrete*” (JGJ55-2011 [35]), the amount of each material in 1 m^3^ concrete was C (cement):W (water):S (sand):G (gravel) = 452 kg:192 kg:526.8 kg:1229.2 kg. To solve the problem of additional water absorption, the method of pre-wetting was used in this test to deal with the water–cement ratio problem, i.e., the recycled aggregates were wetted in advance before the concrete was mixed. Recycled aggregates were set as 0%, 50%, and 100% (equal mass replacement of natural gravel), and rubber granules were set as 0%, 5%, 10%, 15% and 20% (equal volume replacement of sand). Acid washing modification and alkali washing modification of rubber particles formulated modified rubber recycled concrete. Referring to Zhao (2017) [36], R represents recycled aggregate incorporated, K represents rubber particles incorporated, W stands for rubber washed, J represents rubber alkaline washed, and A stands for rubber acid washed. The test mix proportions are shown in Table 3 (the ratios for the other three modifications are the same as in Table 3). RR stands for recycled concrete, ARR represents acid-washed rubber recycled concrete and JRR stands for alkaline-washed rubber recycled concrete.

According to the provisions of “*Standard for test methods of mechanical properties of ordinary concrete*” (GB/T 50081-2002 [37]) and “*Test code for hydraulic concrete*” (SL 352-2006 [38]), the basic mechanical properties were determined by making concrete specimen blocks of the corresponding specifications and sizes. The specimen size for the compressive strength and the splitting tensile strength was 100 × 100 × 100 mm, and the concrete members for the axial compressive strength and the modulus of elasticity were 150 × 150 × 300 mm. The main tests conducted in this paper were the compressive strength test, the splitting tensile strength test, the flexural strength test, the axial compressive strength test, and elastic modulus test. This test used the WAW-300 microcomputer-controlled electro-hydraulic servo universal testing machine produced by Jinan Trial Gold Group Co. (Jinan, China). The maximum range of the test force is 30 T and the displacement resolution is 0.01 mm. This testing machine was used to study the basic mechanical properties of modified rubber recycled concrete, as shown in Figure 2. The axial compressive strength and the compressive strength are two different indicators that describe the compressive properties of an object. The axial compressive strength is a structural performance indicator that takes into account the stability of an object and its ability to resist external loads. The compressive strength, on the other hand, is a purely physical property indicator, which only takes into account the material’s own ability to resist compression.

## 3. Results and Discussion

### 3.1. Basic Mechanical Properties of Rubber Recycled Concrete

#### 3.1.1. The Compressive Strength

Figure 3 shows the relationship between the compressive strength of rubber recycled concrete (RRC) and rubber admixture. From Figure 3, the compressive strength of RRC was inversely proportional to the admixture of recycled aggregates. Compared to normal concrete (R0), the compressive strength was reduced by 6.60% and 10.31% at 50% and 100% recycled aggregate substitution rates, respectively. Further, the compressive strength of RRC was inversely proportional to the rubber admixture. Compared to R0, the 28 d compressive strength was reduced by 6.80%, 9.90%, 16.08%, and 25.36% when the rubber particles were mixed at 5%, 10%, 15%, and 20%, respectively. The reduction in the compressive strength of rubber granules doped with 20% was the greatest, with a maximum of 27.36%.

Both the addition of recycled coarse aggregate and rubber reduced the compressive strength of concrete. This was mainly due to a large number of microcracks within the recycled aggregate crushing process, and the surface was rough, angular, and had lower strength than natural aggregates. Also, this was mainly due to the low modulus of elasticity and the low compressive strength of rubber granules compared to natural aggregates such as sand and stone. In a word, the effects of recycled aggregate and rubber admixture on the compressive strength of concrete were on the large side. The effect of rubber admixture was greater than that of recycled aggregate substitution rate, which was consistent with the results of Zhao (2017) [36] and Zhang (2017) [39].

#### 3.1.2. The Splitting Tensile Strength

Figure 4 shows the relationship between the splitting tensile strength of RRC and rubber admixture. From Figure 4, the splitting tensile strength of RRC decreased gradually with the increase in the admixture of recycled aggregate and rubber particles. Compared to normal concrete (R0), the splitting tensile strength was reduced by 3.70% and 4.81% at 50% and 100% recycled aggregate substitution rates, respectively. Compared with R0, the 28 d splitting tensile strength was reduced by 5.56%, 8.51%, 15.19%, and 21.11% when the rubber particles were mixed at 5%, 10%, 15%, and 20%, respectively. 

Both the addition of recycled coarse aggregate and rubber reduced the splitting tensile strength of concrete. This was because the recycled aggregate was wrapped around the old cement mortar, weakening the bonding ability between the recycled aggregate and the freshly mixed cement mortar. Also, this was due to the hydrophobicity of the rubber granules and the presence of chemicals on the surface of the rubber granules resulting in poor bonding of the rubber granules to the cement mortar. In addition, the reduction in the splitting tensile strength of rubber granules dosed with 20% was the greatest when the recycled aggregate substitution rate was 50% and 100%, with a maximum of 23.08% and 27.24%, respectively. Therefore, the effect of rubber particle admixture on the splitting tensile strength was greater than that of recycled aggregate replacement rate.

#### 3.1.3. The Flexural Strength

Figure 5 shows the relationship between the flexural strength of RRC and rubber admixture. From Figure 5, the flexural strength was negatively correlated with the admixture of recycled aggregates and rubber particles. Compared to R0, the flexural strength was reduced by 5.77% and 7.12% at 50% and 100% recycled aggregate replacement, respectively. The flexural strength was reduced by 3.65%, 8.46%, 13.85%, and 14.62% when the rubber particles were mixed at 5%, 10%, 15%, and 20%, respectively. 

Both the addition of recycled coarse aggregate and rubber reduced the flexural strength of RRC. This was mainly due to the high susceptibility to cracking when tensile stresses were applied at the bond interface between the recycled aggregate and the cement mortar. Also, this was mainly due to the poor bonding of rubber granules to cement mortar. In addition, the reduction in the flexural strength of rubber granules dosed with 20% was the greatest when the recycled aggregate substitution rate was 50% and 100%, with a maximum of 15.10% and 15.32%, respectively. Therefore, rubber granule admixture and recycled aggregate replacement rate affected the flexural strength to approximately the same extent.

#### 3.1.4. The Axial Compressive Strength

Figure 6 shows the relationship between the axial compressive strength and rubber admixture. From Figure 6, the axial compressive strength was negatively correlated with the admixture of recycled aggregates and rubber particles. Compared to R0, the axial compressive strength was reduced by 5.78% and 7.23% at 50% and 100% recycled aggregate replacement, respectively. The axial compressive strength was reduced by 3.37%, 9.64%, 13.98%, and 24.82% when the rubber particles were mixed at 5%, 10%, 15%, and 20%, respectively. In addition, the axial compressive strength of rubber granules doped with 20% was reduced by 21.99% and 21.82% when the recycled aggregate substitution rate was 50% and 100%, respectively. 

From the test phenomenon, the axial compressive concrete specimens mixed with rubber particles showed obvious brittle damage. And the ductility of the specimens increased with the increase in rubber admixture. Therefore, the incorporation of rubber particles enhanced the ductility of concrete, which was in agreement with the findings of Karunarathna et al. (2021) [25]. Therefore, the rubber granule admixture and recycled aggregate replacement rate affected the axial compressive strength of concrete to approximately the same extent.

#### 3.1.5. The Modulus of Elasticity

Figure 7 shows the relationship between the modulus of elasticity and rubber admixture. From Figure 7, the incorporation of both recycled aggregates and rubber particles reduced the modulus of elasticity of concrete. Compared to R0, the modulus of elasticity of concrete for R50 and R100 was reduced by 6.65% and 7.67%, respectively. This was because the particle compactness and the modulus of elasticity of recycled aggregates were lower than that of natural crushed stone. The old cement mortar wrapped around the surface of recycled aggregates weakened the stiffness of the concrete. The modulus of elasticity of concrete was reduced in the range of 3.069–16.62% when rubber particles were mixed at 5–20%. This was because the modulus of elasticity of rubber granules was lower than that of natural sand, stone, and recycled aggregates. Zhao (2017) [36] reported that the addition of rubber powder distributed numerous elastomers in the cement mortar. When cement concrete was subjected to external loading, these elastomers were able to dissipate the generation and development of microcracks, thus making the concrete more elastic but less rigid. 

In addition, the modulus of elasticity of rubber granules doped with 20% decreased by 20.27% and 19.94% when the recycled aggregate substitution rate was 50% and 100%, respectively. Thus, the rubber particle admixture and recycled aggregate replacement rate affected the modulus of elasticity of concrete to approximately the same extent.

### 3.2. Basic Mechanical Properties of Modified Rubber Recycled Concrete

#### 3.2.1. The Compressive Strength

Figure 8 shows the effect of modification methods on the compressive strength of concrete. Overall, water washing, alkaline washing, and acid washing significantly increased the compressive strength of concrete. Compared with the R0, the compressive strength of water-washed, alkaline-washed, and acid-washed modification methods increased by 1.83–4.14%, 6.63–10.77%, and 4.35–9.67%, respectively, when rubber particles were mixed with 5–20%. When the recycled aggregate admixture was certain, the rubber particle admixture was positively correlated with the strength of modified rubber recycled concrete (MRRC). The water-washed, alkaline-washed, and acid-washed modification methods were improved by 2.46–3.48%, 5.67–10.44%, and 3.69–6.01%, respectively for the compressive strength, when the recycled aggregate mixing rate was 100% and the rubber particles mixing rate was 5–20%, respectively. 

In summary, the effects of the three modifications on the compressive strength of MRRC were ranked as follows: alkali washing > acid washing > water washing. And the alkali washing modification method increased the compressive strength by a greater amount, up to 12.39%, which was closer to the scope of studies by He et al. (2016) [30] and Su et al. (2022) [31].

#### 3.2.2. The Splitting Tensile Strength

Figure 9 shows the effect of the modification method on the splitting tensile strength. Overall, water washing, alkaline washing, and acid washing all significantly increased the splitting tensile strength of concrete by an average of 4.64%, 10.46%, and 7.62%, respectively. From Figure 9, compared to R0, the splitting tensile strength of water-washed, alkaline-washed, and acid-washed modification methods increased by 2.35–7.06%, 8.92–12.22%, and 6.88–8.23%, respectively, when the rubber particles were mixed at a dosage of 5–20%. When the recycled aggregate admixture was certain, the rubber particle admixture was positively correlated with the splitting tensile strength of MRRC. The water-washed, alkaline-washed, and acid-washed modification methods were improved by 2.07–4.04%, 5.89–8.57%, and 4.15–5.38%, respectively, when the recycled aggregate mixing rate was 50%. 

In summary, the effects of the three modification methods on the splitting tensile strength of MRRC are in the following order: alkali washing > acid washing > water washing. The alkali washing modification method increased the concrete splitting tensile strength by a greater amount, up to 15.51%.

#### 3.2.3. The Flexural Strength

Figure 10 shows the effect of the modification method on the flexural strength of concrete. Overall, all three modifications significantly increased the flexural strength of MRRC. From Figure 10, the splitting tensile strength of the water-washed, alkaline-washed, and acid-washed modification methods increased by an average of 1.46%, 3.70%, and 2.63%, respectively. When the recycled aggregate admixture was certain, the rubber particle admixture was positively correlated with the flexural strength of MRRC. When the recycled aggregate and rubber particles were mixed with 50% and 5–20%, respectively, the water-washed, alkaline-washed, and acid-washed modification modes were improved by 1.03%, 3.79%, and 2.82%. 

In summary, the three modifications on the flexural strength of MRRC were ranked as follows: alkali washing > acid washing > water washing. The alkali washing modification method had the greatest effect on the flexural strength of concrete, with a maximum of 9.2%.

#### 3.2.4. The Axial Compressive Strength

Figure 11 shows the effect of the three modifications on the axial compressive strength of MRRC. From Figure 11, all three modifications increased the axial compressive strength of MRRC. Compared to the R0, the axial compressive strength of the water-washed, alkaline-washed, and acid-washed modification methods increased by an average of 1.74%, 5.45%, and 3.13%, respectively, when rubber particles were mixed at 5–20%. When the recycled aggregate admixture was certain, the rubber particle admixture was positively correlated with the axial compressive strength of MRRC. The modification methods of rubber washing, rubber alkaline washing, and rubber acid washing were improved by 1.09–3.65%, 2.99–7.47%, and 2.17–4.82%, respectively, when the recycled aggregate and rubber particles were mixed with 100% and 5–20%, respectively.

In summary, the three modifications on the flexural strength of concrete are ranked as follows: alkali washing > acid washing > water washing. The alkali-washing modification method has the greatest effect on the flexural strength of concrete, with a maximum of 7.53%, which was closer to the scope of studies by Zhao (2017) [36].

#### 3.2.5. The Modulus of Elasticity

Figure 12 shows the effect of the three modifications on the modulus of elasticity of concrete. From Figure 12, all three modifications improved the modulus of elasticity of MRRC. Compared to the R0, the modulus of elasticity of the water-washed, alkaline-washed, and acid-washed modification methods increased by 0.53–2.76%, 2.37–6.44%, and 1.58–4.29%, respectively, when the rubber particles were mixed at 5–20%. When the recycled aggregate admixture was certain, the rubber particle admixture was positively correlated with the modulus of elasticity of MRRC. The modification methods of rubber water washing, rubber alkaline washing, and rubber acid washing were improved by an average of 4.57%, 7.45%, and 5.96% when the recycled aggregate and rubber particles were mixed with 100% and 5–20%, respectively.

In summary, the three modification methods on the modulus of elasticity of concrete were ranked as follows: alkali washing > acid washing > water washing [40]. The alkali washing modification method showed the greatest improvement in the modulus of elasticity of concrete, with a maximum of 11.19%.

## 4. Conclusions

Based on indoor tests, the mechanical properties of modified rubber recycled concrete (MRRC) were investigated. The main conclusions follow.

The incorporation of both rubber particles and recycled aggregates reduced the basic mechanical properties of concrete. And the extent of the effect of rubber particles was greater than that of recycled aggregate. The reduction in the compressive strength, the splitting tensile strength, the flexural strength, the axial compressive strength, and the modulus of elasticity of rubber recycled concrete (RRC) was the greatest when recycled aggregates and rubber particles were mixed at 100% and 20%, respectively, with a maximum reduction of 15.32–27.36%.The modification methods of water washing, alkali washing, and acid washing improved the mechanical properties of rubber recycled concrete by 1.09–4.57%, 2.99–10.44%, and 2.17–6.01%, respectively. The effects of the three modification methods on the mechanical properties of concrete were ranked as: alkali washing > acid washing > water washing. The alkali washing modification method improved the compressive strength and the splitting tensile strength more significantly, with a maximum enhancement effect of 12.39% and 15.51%.The sensitivity of the modification method to the mechanical properties of concrete was ranked as follows: splitting tensile strength > compressive strength > modulus of elasticity > flexural strength > axial compressive strength. Compared to the unmodified rubber recycled concrete, the compressive strength, the splitting tensile strength, the flexural strength, the axial compressive strength, and the modulus of elasticity of rubber recycled concrete increased by 12.39%, 15.51%, 9.2%, 7.53%, and 11.19%, respectively, when recycled aggregate and rubber particles were mixed at 100% and 20%, respectively.

## Figures and Tables

**Figure 1 materials-17-02217-f001:**
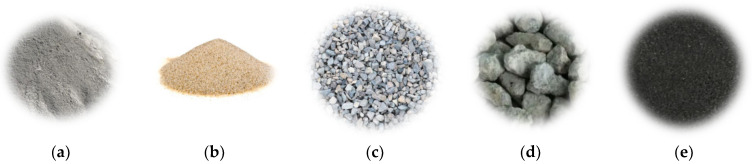
Test materials: (**a**) cement; (**b**) sand; (**c**) coarse aggregate; (**d**) recycled aggregate; (**e**) rubber granules.

**Figure 2 materials-17-02217-f002:**
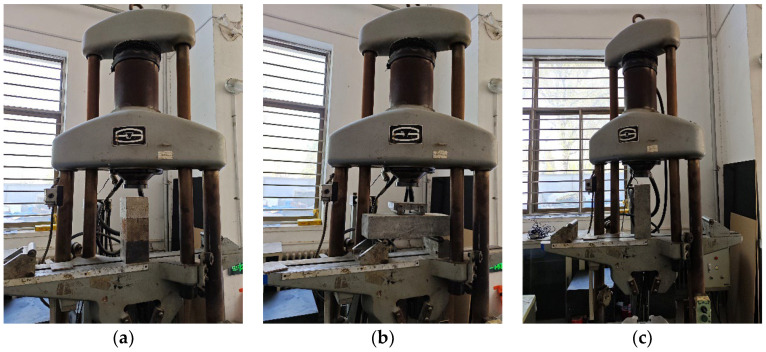
Test photographs of (**a**) the compressive strength; (**b**) the flexural strength; (**c**) the modulus of elasticity.

**Figure 3 materials-17-02217-f003:**
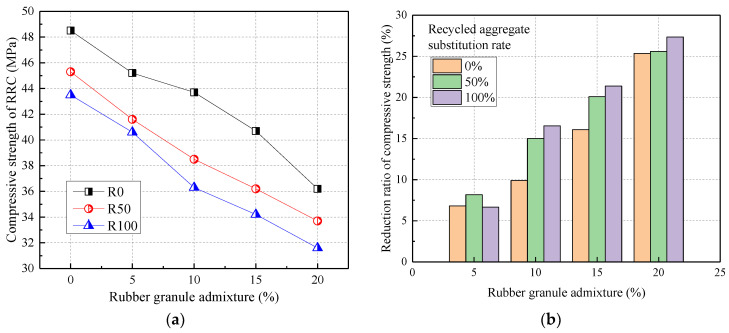
The compressive strength of RRC: (**a**) the effect of rubber granule admixture; (**b**) the reduction ratio.

**Figure 4 materials-17-02217-f004:**
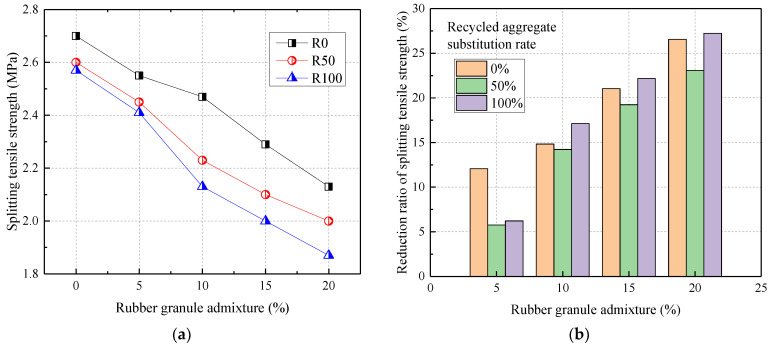
The splitting tensile strength of RRC: (**a**) the effect of rubber granule admixture; (**b**) the reduction ratio.

**Figure 5 materials-17-02217-f005:**
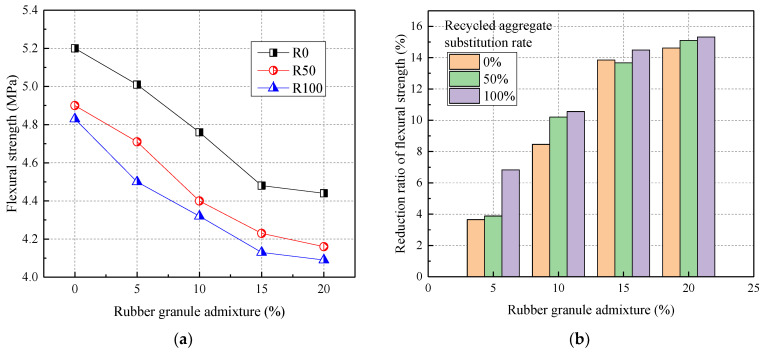
The flexural strength of RRC: (**a**) the effect of rubber granule admixture; (**b**) the reduction ratio.

**Figure 6 materials-17-02217-f006:**
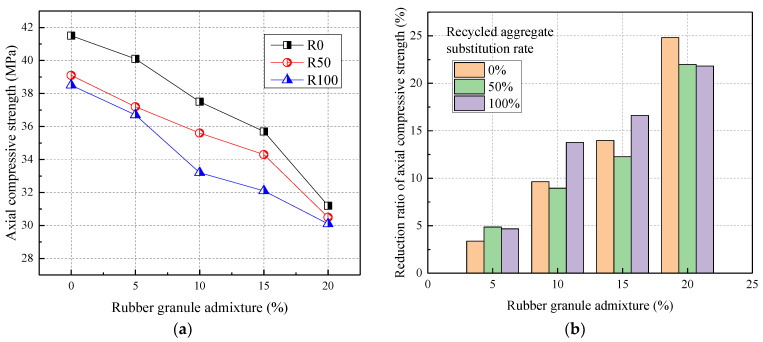
The axial compressive strength of RRC: (**a**) the effect of rubber granule admixture; (**b**) the reduction ratio.

**Figure 7 materials-17-02217-f007:**
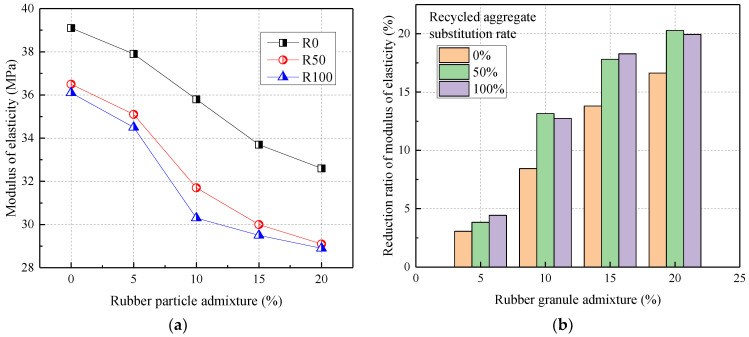
The modulus of elasticity of RRC: (**a**) the effect of rubber particle admixture; (**b**) the reduction ratio.

**Figure 8 materials-17-02217-f008:**
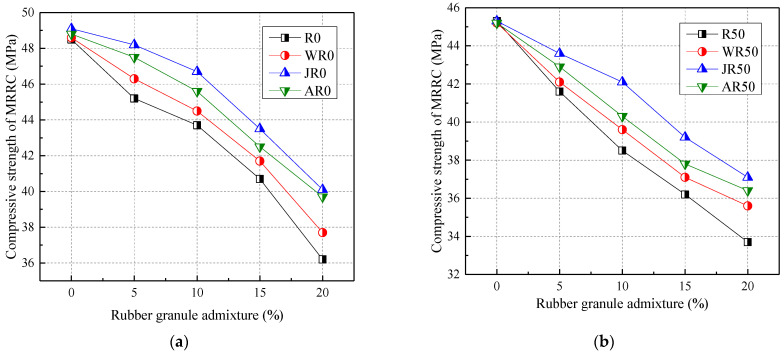
Effect of modification methods on the compressive strength: recycled aggregate substitution rate of (**a**) 0%; (**b**) 50%; (**c**) 100%; (**d**) the percentage increase in the compressive strength.

**Figure 9 materials-17-02217-f009:**
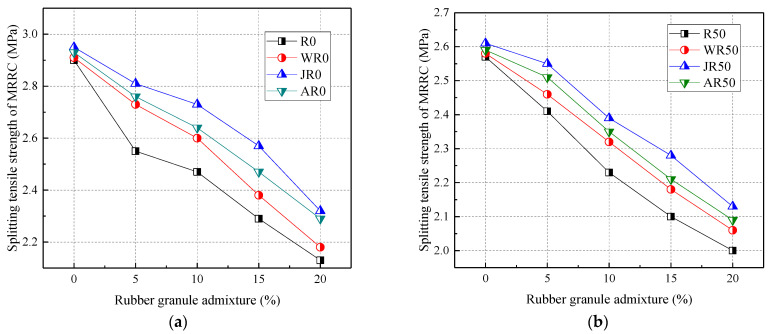
Effect of modification method on the splitting tensile strength: recycled aggregate substitution rate of (**a**) 0%; (**b**) 50%; (**c**) 100%; (**d**) the percentage increase in the splitting tensile strength.

**Figure 10 materials-17-02217-f010:**
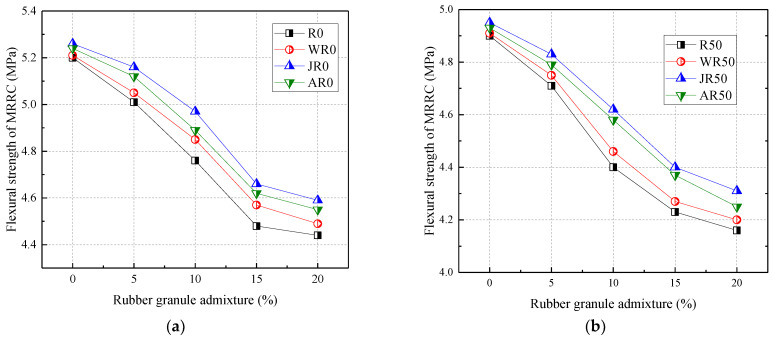
Effect of modification methods on the flexural strength: recycled aggregate substitution rate of (**a**) 0%; (**b**) 50%; (**c**) 100%; (**d**) the percentage increase in the flexural strength.

**Figure 11 materials-17-02217-f011:**
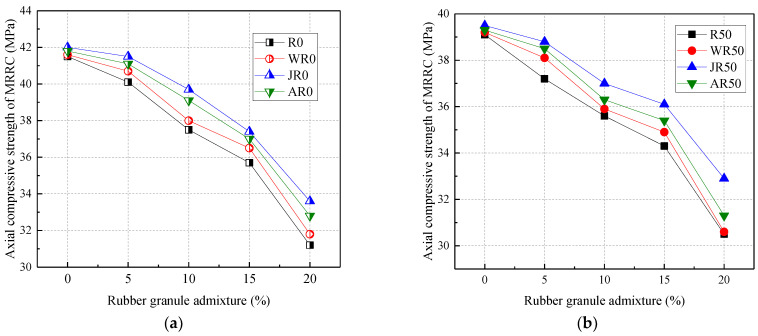
Effect of modification methods on the axial compressive strength: recycled aggregate substitution rate of (**a**) 0%; (**b**) 50%; (**c**) 100%; (**d**) the percentage increase in the axial compressive strength.

**Figure 12 materials-17-02217-f012:**
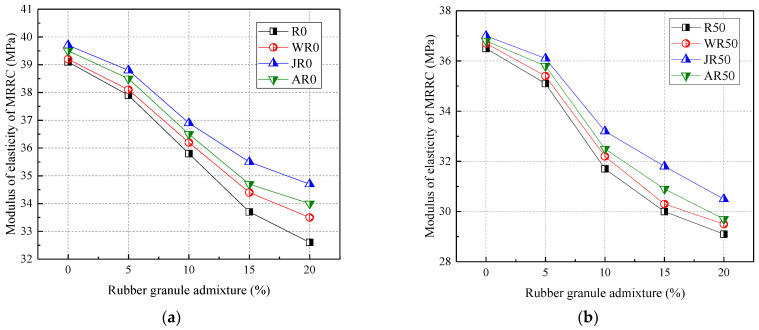
Effect of modification method on the modulus of elasticity: recycled aggregate substitution rate of (**a**) 0%; (**b**) 50%; (**c**) 100; (**d**) the percentage increase in the modulus of elasticity.

**Table 1 materials-17-02217-t001:** Basic performance indexes of cement.

Density (kg/m^3^)	Initial Setting Time (min)	Final Setting Time (min)	Compressive Strength (MPa)		Flexural Strength (MPa)	
			3 d	28 d	3 d	28 d
3050	200	265	27.5	49.3	5.5	8.0

**Table 3 materials-17-02217-t003:** Mix proportions.

Item	Mix Proportions of Cubic Concrete (kg/m^3^)					
	Water	Cement	Sand	Rubber Granule	Natural Crushed Stone	Recycled Aggregate
R0-K0	192	452	526.8	0	1229.2	0
R0-K5			500.46	26.34		
R0-K10			474.12	52.68		
R0-K15			447.78	79.02		
R0-K20			421.44	105.36		
R50-K0	192	452	526.8	0	614.6	565.4
R50-K5			500.46	26.34		
R50-K10			474.12	52.68		
R50-K15			447.78	79.02		
R50-K20			421.44	105.36		
R100-K0	192	452	526.8	0	0	1130.8
R100-K5			500.46	26.34		
R100-K10			474.12	52.68		
R100-K15			447.78	79.02		
R100-K20			421.44	105.36		

**Table 2 materials-17-02217-t002:** Basic performance indicators for coarse and fine aggregates.

Types		Particle Size /(mm)	Bulk Density /(kg/m^3^)	Apparent Density /(kg/m^3^)	Water Absorption /%	Mud Content /%	Indicators of Crushing /%
Coarse aggregate	Natural coarse aggregate	5–25	2580	1340	0.8	0.45–0.47	10
	Recycled coarse aggregate	5–25	2500	1210	9.5	0.65–0.68	16.2
Fine aggregate	Sand	2–4	1580	2530	/	/	/
	Rubber granule	2–5	/	1210	/	/	/

## Data Availability

Data will be made available on request.

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
