# Peer review of "Effect of the Inorganic Modification Mode on the Mechanical Properties of Rubber Recycled Concrete"

_materials, 2024, doi:10.3390/ma17102217_

Round 1

Reviewer 1 Report

Comments and Suggestions for Authors

Dear authors,

thank you for ypur work. 

You cn make Introduction deeper, for example

Rassokhin A.S., Ponomarev A.N., Karlina A.I. High-performance fine-grained nanostructured concrete based on low strength aggregates // Magazine of Civil Engineering, 2022, 114(6), 11413

In Table 2 please write the range for all measurements, not only for Particle size.

Give more information about testing equipment.

Give comments afret fig.1, fig.2, fig.3, fig.4, fig.5, fig.6, fig.7 etc. Don't finish paragraph with fig. Give analysis.

In Results and Discussion now you have mix of them. Better to fix results and next give discussion.

Best regards, reviewer

Author Response

Thank you very much for your letter. The reviewers’ comments concerning our manuscript entitled “Effect of inorganic modification mode on mechanical properties of rubber recycled concrete”. These comments are all valuable and very helpful for revising and improving our paper, as well as important guiding significance to our research. We have studied comments carefully and have made corrections which we hope meet with approval. All changes are marked in the track changes version. The main corrections in the paper and the responses to the reviewer’s comments are as follows.

Reviewer 2 Report

Comments and Suggestions for Authors

This article targets the effect of modification mode on the mechanical properties of rubber-recycled concrete. The detailed comments are as follows:

1. No need to show (1), (2) and (3) in the abstract.

2. Again no need to provide numbering ((1), (2), (3), (4)) in the introduction section. 

3. Introduction section needs revision. It should only discuss the literature related to current work. No need to discuss FRP-related works. Moreover, the focus should be on articles that have both recycled concrete and waste rubber. Authors should also highlight the latest techniques for improving the performance of RRC such as compression casting technique. Only discuss the results from literature related to the current study.

4. The water absorption of recycled aggregates is different than natural aggregates. Moreover, water absorptions of rubber and sand are also different. How authors have maintained the constant water-to-cement ratio.

5. Mention the test standrads followed during this work.

6. The difference between compressive strength and axial compressive strength is unclear.

7. Some images of the test set-up should be included. Furthermore, the images of waste rubber before and after modifications should be provided.

8. The impurity level of recycled aggregates should be mentioned.

9. The results are well-explained. However, proof-reading is needed.

10. Microstructure images should be provided to support the results.

Comments on the Quality of English Language

Overall, the article is well-written concerning the quality of language. However, proofreading is needed to further improve the structure of sentences.

Author Response

(The authors gave the same response as above.)

Reviewer 3 Report

Comments and Suggestions for Authors

The text needs a general review from the title, summary, introduction, and methodology.

1. Effect of modification mode...? What modification? The physical modification involves including mechanical grinding and shaping, heat treatment, and microwave or electric pulse treatment. Chemical modification includes acid treatment removal, water glass strengthening, carbonation strengthening, inorganic slurry strengthening, and polymer strengthening. Microbial modification consists of using specific microorganisms that induce carbon deposition modification.

2. The word modification is used in two situations: “Modified rubber recycled concrete (MRRC) is a new type of material that combines the properties of rubber granules from waste tires with recycled building concrete materials.” “For modification methods, He et al. (2016) and Su et al. (2022) investigated the effects of three surface modifications (oxidation,

sulfonation, and alkalization) on the mechanical properties o.. .."

In another item: “Besides, the research on the effect of different modification methods

on the mechanical properties of MRRC is little and needs to be further explored.”

  3. The aim of the research is the effect of different modification methods

on the mechanical properties of MRRC. The methodology does not describe the treatment and who it was applied to (only mentions)

4. Explain the difference between the tests performed: “ compressive strength, ......... axial compressive strength “

5. Resistance or strength? (figures)

5. It is interesting to analyze the results against the mechanisms that cause effects in properties. The study of porosity is an essential aspect of such analysis.

Author Response

(The authors gave the same response as above.)

Round 2

Reviewer 2 Report

Comments and Suggestions for Authors

The authors have revised the article based on the reviewer's comments.